# Validating Methylated HOXA9 in Bronchial Lavage as a Diagnostic Tool in Patients Suspected of Lung Cancer

**DOI:** 10.3390/cancers13164223

**Published:** 2021-08-22

**Authors:** Sara W. C. Wen, Rikke F. Andersen, Kristian Rasmussen, Caroline Brenner Thomsen, Torben Frøstrup Hansen, Line Nederby, Henrik Hager, Anders Jakobsen, Ole Hilberg

**Affiliations:** 1Department of Oncology, Vejle Hospital, University Hospital of Southern Denmark, 7100 Vejle, Denmark; Caroline.Emilie.Brenner.Thomsen@rsyd.dk (C.B.T.); Torben.Hansen@rsyd.dk (T.F.H.); Anders.Jakobsen@rsyd.dk (A.J.); 2Institute of Regional Health Research, University of Southern Denmark, 5230 Odense M, Denmark; henrik.hager@rsyd.dk (H.H.); ole.hilberg@rsyd.dk (O.H.); 3Department of Clinical Biochemistry, Vejle Hospital, University Hospital of Southern Denmark, 7100 Vejle, Denmark; rikke.fredslund.andersen@rsyd.dk (R.F.A.); line.nederby@rsyd.dk (L.N.); 4Department of Internal Medicine, Vejle Hospital, University Hospital of Southern Denmark, 7100 Vejle, Denmark; kristian.rasmussen@rsyd.dk; 5Department of Pathology, Vejle Hospital, University Hospital of Southern Denmark, 7100 Vejle, Denmark

**Keywords:** lung cancer, circulating tumor DNA, methylated HOXA9

## Abstract

**Simple Summary:**

Diagnosing lung cancer requires invasive procedures with high risk of complications. Methylated tumor-specific DNA has been suggested as a biomarker for lung cancer. The present study aimed to develop and validate the biomarker methylated *HOXA9* in fluid from the lung collected during bronchoscopy. This biomarker has a clinically relevant sensitivity and specificity for the diagnosis of lung cancer. Future research should focus on determining the optimal combination of biomarker and biologic specimen.

**Abstract:**

Diagnosing lung cancer requires invasive procedures with high risk of complications. Methylated tumor DNA in bronchial lavage has previously shown potential as a diagnostic biomarker. We aimed to develop and validate methylated *HOXA9* in bronchial lavage as a diagnostic biomarker of lung cancer. Participants were referred on suspicion of lung cancer. Ten mL lavage fluid was collected at bronchoscopy for analysis of methylated *HOXA9* based on droplet digital PCR according to our previously published method. *HOXA9* status was compared with the final diagnosis. The Discovery and Validation cohorts consisted of 101 and 95 consecutively enrolled participants, respectively. In the discovery cohort, the sensitivity and specificity were 73.1% (95% CI 60.9–83.2%) and 85.3% (95% CI 68.9–95.0%), respectively. In the validation cohort, the values were 80.0% (95% CI 66.3–90.0%) and 75.6% (95% CI 60.5–87.1%), respectively. A multiple logistic regression model including age, smoking status, and methylated *HOXA9* status resulted in an AUC of 84.9% (95% CI 77.3–92.4%) and 85.9% (95% CI 78.4–93.4%) for the Discovery and Validation cohorts, respectively. Methylated *HOXA9* in bronchial lavage holds potential as a supplementary tool in the diagnosis of lung cancer with a clinically relevant sensitivity and specificity. It remained significant when adjusting for age and smoking status.

## 1. Introduction

Lung cancer is considered the deadliest cancer worldwide [1] and is often diagnosed at a late stage [2,3,4]. Screening with low-dose computed tomography (CT) scans is recommended in the USA and some European countries based on the results from large screening trials, including the US-based National Lung Screening Trial (NLST) [5] and the Dutch-Belgian Nederlands-Leuvens Longkanker Screenings Onderzoek (NELSON) [6]. These studies concluded that there was a reduction in mortality of at least 20% in the CT-screened cohort. The Danish lung cancer screening trial found no significant reduction in mortality but reported a significantly larger fraction of early-stage cancers in the screening group [7].

CT scans produce a large number of false positive findings. The NLST reported a false-positive rate of 26.6% [5]. The NELSON trial reported a false-positive rate of 1.2% but approximately 20% of all tests at baseline were deemed indeterminate and required a repeated CT scan [6]. According to the Lung-RADS guidelines [8], patients with solid nodules ≥8 mm should be followed up with CT scans every three months. Such an initiative would cause great strain on departments of radiology and pulmonary medicine, and a screening program in general represents a significant economic burden.

Additional diagnostic criteria to help confirm or reject a cancer diagnosis could improve cost-benefit and reduce workload [9,10]. Risk prediction models, such as the PLCO_M2012_, showed better sensitivity without loss of specificity compared to the NLST selection criteria [11]. Risk prediction models could be further improved by including biomarkers [12].

Circulating tumor-specific DNA (ctDNA) has been suggested as a promising biomarker for diagnosing lung cancer. However, studies have indicated that small or early-stage lung tumors do not shed as much DNA into the circulation as larger tumors and hence are more difficult to detect in the blood [13,14]. This could be overcome by using material collected closer to the tumor site, as this material likely contains more tumor DNA. This idea was indicated in previous studies comparing tumor DNA detected in blood and sputum [15,16] to tumor DNA detected in bronchial washings [17].

Aberrant methylation of the promoter region can affect gene expression and has been linked to cancer development [18]. Hypermethylation usually inhibits gene transcription, while hypomethylation increases transcription [18]. The *homeobox A9* (*HOXA9*) gene encodes a DNA-binding transcription factor [19]. *HOXA9* has been shown to be dysregulated in many solid tumors, including lung cancer [20], and *in vitro* experiments have found downregulation of *HOXA9* to enhance migratory potential [21] and stimulate cell invasiveness [22]. Hypermethylated *HOXA9* has been suggested as a diagnostic biomarker in lung cancer [15,23], and our group has shown that hypermethylated *HOXA9* is a negative prognostic biomarker in advanced lung cancer [24].

The objectives of this study were to develop and validate the use of methylated *HOXA9* in bronchial lavage fluid as a diagnostic biomarker of lung cancer. We hypothesize that methylated *HOXA9* in bronchial lavage fluid can serve as a valuable adjunct in the diagnosis of lung cancer.

We conclude that methylated *HOXA9* in bronchial lavage holds potential as a supplementary tool in the diagnosis of lung cancer because it has a clinically relevant sensitivity and specificity.

## 2. Materials and Methods

### 2.1. Participants and Study Design

Participants were prospectively enrolled in this observational study at the Department of Medicine, Vejle Hospital, University Hospital of Southern Denmark, from October 2018 to December 2019. The first 107 participants were allocated to the Discovery cohort, and the subsequent 100 participants were allocated to the Validation cohort. Inclusion criteria were referral for diagnostic work-up, including bronchoscopy, on suspicion of lung cancer and age > 18 years. The exclusion criterion was severe comorbidity preventing the participant from completing the planned follow-up procedures. For the Validation cohort, a previous diagnosis of lung cancer was introduced as an additional exclusion criterion. Participants were followed for at least six months after enrolment.

The study was performed according to the Declaration of Helsinki of 1975 and the Danish data protection legislation. All participants provided written, informed consent before enrolling in the study. The study protocol was approved by the Regional Committee for Health Research Ethics of Southern Denmark (No. S-20180052, 22 June 2018).

### 2.2. Definition of Patient Characteristics

Employment status was evaluated from the medical record. If the participant was older than 65 years, he/she was considered retired unless otherwise specified in the medical record. Smoking status was categorized as never if the participant had smoked less than one pack year in his/her lifetime and as ever if he/she had smoked more than one pack year. One pack year was defined as 20 cigarettes per day for a year or the equivalent in other tobacco types. Performance status was categorized as defined by the Eastern Cooperative Oncology Group. FEV1 was the forced expiratory volume of air in 1 s recorded in liter. Any comorbidity was defined as the participant having any condition that required regular medication. Cancer within five years was defined as any diagnosis of malignancy within the past five years excepting non-melanoma skin cancer and carcinoma in situ cervix uteri, while previous lung cancer was defined as any previous diagnosis of lung cancer. All clinical participant characteristics were recorded at the first doctor’s appointment.

### 2.3. Diagnostic Work-Up

The standard diagnostic work-up on suspicion of lung cancer consisted of a CT scan of the chest and abdomen and, depending on the result, further investigations were initiated. These included: blood samples, full-body positron-emission tomography (PET) CT scan, bronchoscopy with endobronchial ultrasound (EBUS)-guided fine-needle aspiration, or CT-guided transthoracic needle aspiration. The results of these investigations were evaluated by a multidisciplinary tumor board consisting of doctors specializing in pulmonary medicine, radiology, nuclear medicine, pathology, thoracic surgery, and clinical oncology.

### 2.4. Bronchoscopy and Bronchial Lavage Sampling

Bronchoscopy was performed under general anesthesia with either a BF-1TH190 or BF-H190 Olympus video bronchoscope (Olympus, Shinjuku City, Tokyo, Japan). The bronchoscope was introduced through the tracheal tube. In the case of a visible lesion, the bronchoscope was positioned as closely as possible to the tumor, and, subsequently, 10 mL of sterile saline was instilled and retrieved. In the case of no visible lesion, the bronchoscope was positioned as closely as possible to the tumor site, as identified on the CT scan, and bronchial lavage was performed. The project samples were collected after lavage samples had been taken for cytological and microbiological examination. Biopsies were taken either directly from visible lesions or guided by endobronchial ultrasound.

We aimed to collect a sample volume of 10 mL because this is the standard volume used for clinical examinations. The recovery of bronchial lavage fluid ranged from 3–7 mL.

### 2.5. Reference Test

The gold standard reference in this study was a histopathology-confirmed diagnosis of lung cancer, as agreed upon by the tumor board. Patients with a histopathology-confirmed diagnosis of malignancy other than lung cancer were excluded from this study (Figure 1). Patients with a suspicious nodule on CT scan were categorized as controls if they did not have histopathology-confirmed lung cancer after six months.

### 2.6. Analysis of Methylated HOXA9

Bronchial lavage fluid was centrifuged at 300× *g* for 10 min, and the supernatant was frozen at −80 °C pending further analysis. DNA was extracted from 2 mL bronchial lavage fluid with the DSP Circulating DNA kit (Qiagen, Hilden, Germany) as recommended by the manufacturer. The method of analysis of methylated *HOXA9* has been published previously [25,26,27]. Briefly, DNA was bisulfite converted using the EZ DNA Methylation-Lightning Kit (Zymo Research, Irvine, CA, USA) according to the manufacturer’s instructions. The bisulfite-converted DNA was analyzed for methylated *HOXA9* by an in-house methylation-specific droplet digital polymerase chain reaction (ddPCR) assay and read on a QX100 Droplet Digital Reader (Bio-Rad, Hercules, CA, USA). Several checkpoints ensured optimal performance of the assay. Spike-in of *CPP1* served as an internal control of DNA extraction efficacy [28], and the ¦*Beta2 microglobulin* gene was used as a surrogate for the total amount of cell-free DNA before bisulfite conversion. This checkpoint was a quality control of the DNA extraction step. The methylation-specific ddPCR assay included water as a negative control, a pool of lymphocyte DNA from healthy donors as a non-cancer control, and Universal Methylated DNA Standard (Zymo Research, Irvine, CA, USA) as a methylated control. The primer and probe sequences can be found in an online data supplement. Analysis of methylated *HOXA9* was performed blinded to the clinical endpoint.

### 2.7. Determining the Optimal Cut-Off for Methylated HOXA9

A volume of 2 mL bronchial lavage fluid had the best performance regarding cell-free DNA yield and the least degree of PCR inhibition.

The limit of blank was set at ≤ 4 droplets containing methylated HOXA9, based on data from 50 healthy donors in a plasma-based assay which was validated on another 50 donors [27]. Methylated *HOXA9* was normalized to the *albumin* gene [29] using the formula:(Methylated *HOXA9* copies/*albumin* copies) * 100(1)
for all samples with ≥ 5 methylated *HOXA9* containing droplets. Methylated *HOXA9* was normalized to albumin to diminish the effect of the total cell-free DNA level on the results. The levels of methylated *HOXA9* are illustrated in the Appendix A. The normalized values were used to establish the optimal cut-off in a receiver operating characteristics (ROC) analysis of the Discovery cohort. The area under the curve (AUC) was 81.5% (95% CI 73.9–89.1%). The optimal cut-off was ≥0.13%, which resulted in 77.2% correctly classified samples and a sensitivity and specificity of 73.1% and 85.3%, respectively. The cut-off was chosen to represent the level of methylated *HOXA9* which resulted in the highest number of correctly classified samples (true positive plus true negative samples). This cut-off was then applied to the Validation cohort and used to dichotomize methylated *HOXA9* for use in the statistical analyses.

Participants with methylated *HOXA9* ≥ 0.13% were considered *HOXA9*-positive in the statistical analyses, while participants with methylated *HOXA9* < 0.13% were considered *HOXA9*-negative.

### 2.8. Statistical Analysis

Categorical variables were reported as fractions (percentages), and continuous variables were reported as median and interquartile ranges (IQR). The Chi-squared test and Fisher’s exact test were used to compare categorical values as appropriate. The Wilcoxon rank-sum test was used to test associations between continuous variables. Sensitivity, specificity, positive predictive value (PPV) and negative predictive value (NPV) were calculated for methylated *HOXA9* status as a binary biomarker. A multiple logistic regression model was developed using smoking status (never vs. ever smoker) and age as clinical predictors and methylated *HOXA9* status as a biomarker. The prediction models were depicted by ROC curves. The models were developed on data from the Discovery cohort and then fitted to data from the Validation cohort. All tests were two-sided; *p*-values < 0.05 were considered statistically significant. All analyses were performed using STATA 16IC (StataCorp LLC, College Station, TX, USA).

## 3. Results

### 3.1. Participant Characteristics

Participants were consecutively and prospectively enrolled from October 2018 to December 2019. The Discovery and Validation cohorts included 101 and 95 participants, respectively, as illustrated in Figure 1. Methylated *HOXA9* status for the patients with other cancers is reported in the online data Appendix A. The participant characteristics for cases and controls in both cohorts are summarized in Table 1. Generally, cases were older, less likely to be employed, had greater tobacco consumption and poorer lung function. Note that patients with a previous lung cancer diagnosis were included in the Discovery cohort but not in the Validation cohort.

Tumor characteristics and histopathology are reported in Table 2. Not all participants had a visible tumor on the CT scan but were referred for further examination based on symptoms, e.g., hemoptysis. Unsurprisingly, cases tended to have longer tumor diameters than controls; adenocarcinomas comprised the largest proportion of the confirmed lung cancers followed by squamous cell carcinomas, and more than half of patients were diagnosed at stages 3 or 4.

### 3.2. Methylated HOXA9 and Lung Cancer

Methylated *HOXA9* measured on bronchial lavage fluid was used as a binary diagnostic biomarker. The full range of diagnostic measures are reported in Table 3. Generally, this biomarker showed better specificity than sensitivity in the Discovery cohort, while the opposite was true for the Validation cohort. However, the confidence intervals overlapped considerably. The diagnostic measures for the Discovery cohort with previous lung cancers excluded can be viewed in the Appendix A.

### 3.3. Predictive Modelling

The prediction model was developed based on data from the Discovery cohort. We included only age and smoking status as clinical predictors and methylated *HOXA9* status as a biomarker, given that there were only 34 controls in that cohort. The two clinical markers were chosen because they are the most commonly used criteria when selecting participants for lung cancer screening. Univariate logistic regression analyses performed on all participant characteristics can be seen in the online data Appendix A. In the Discovery cohort, the clinical regression model had an AUC of 66.6% (95% CI 55.6–78.7%), while the model which included methylated *HOXA9* status had an AUC of 84.9% (95% CI 77.3–92.4%, *p* < 0.001). In the Validation cohort, the AUCs were 71.7% (95% CI 61.4–82.1%) and 85.9% (95% CI 78.4–93.4%, *p* = 0.003), respectively, for the models with and without *HOXA9* status. Please refer to the online data supplement for further information regarding the regression models and for a model on the Discovery cohort excluding previous lung cancers (Appendix A). The models were visualized by ROC curves (Figure 2). 

A multiple logistic regression model showed a statistically significant diagnostic impact of methylated *HOXA9* when adjusting for age and smoking status (Table 4). Further information about the models can be found in the Appendix A.

## 4. Discussion

Low-dose CT-based screening for lung cancer is likely to be introduced in Denmark in the near future, resulting in an increased workload for hospital-based health professionals and many false positive or indeterminate findings. Biomarkers could be used to improve risk assessment before CT-based screening [12]. In the present biomarker validation study, we found that detectable methylated *HOXA9* in bronchial lavage fluid had a clinically relevant sensitivity and specificity. The biomarker continued to have diagnostic impact after adjusting for age and smoking status.

These results are in line with findings by Roncarati et al. [17], who reported a sensitivity and specificity of 97% and 74%, respectively, for a four-gene biomarker panel analyzed on bronchial washings. That study and the present study are similar regarding clinicopathological features and methylation analysis method. The higher sensitivity reported by Roncarati et al. could be due to the four-gene panel, as additional markers have previously shown to increase sensitivity, albeit at the cost of specificity [30]. Roncarati et al. used the cell pellets from bronchial washings for DNA purification, while we used the supernatant. This could also cause differences in diagnostic impact.

In contrast, Villalba et al. [31] reported a sensitivity and specificity of 52% and 91%, respectively, for hypomethylation of *transmembrane serine protease 4* (*TMPRSS4*) in bronchoalveolar lavage fluid from stages I–II non-small cell lung cancers. They observed no significant differences when considering all stages. This likely reflects the difference between the genes investigated, as the studies were similar in most other regards. *TMPRSS4* encodes a membrane-bound serine protease with an unknown function [32], while *HOXA9* encodes a transcription factor. These genes may differ with respect to when and how aberrant methylation develops during oncogenic transformation. Such potential differences are likely to explain the discrepancy between the studies.

We found that methylated *HOXA9* status had a moderate sensitivity and a high specificity in the Discovery cohort, while this was somewhat reversed in the Validation cohort. Hence, the Discovery cohort had more false negative results, and the Validation cohort had more false positive results. Peripheral tumors are more difficult to visualize with a bronchoscope, and the bronchial lavage may be performed some distance from the tumor. This could result in false negative test results. The Validation cohort had a false positive rate of more than 20%. False positive results were evenly distributed among patients with cryptogenic organizing pneumonia, granulomatous inflammation and acute inflammatory disease. They could represent patients with pre-malignant lesions or lesions which would spontaneously resolve over time. A study by Wong Doo et al. [33] suggested that aberrant DNA methylation in blood is present years before a diagnosis of mature B-cell neoplasm is confirmed. This supports the idea that hypermethylation of *HOXA9* is an early event which may or may not lead to malignant transformation. Closer monitoring could be a possible implication of detecting hypermethylated *HOXA9* in bronchial lavage from a patient during diagnostic workup. *HOXA9* has previously been shown to be downregulated in response to inflammatory signals [34], which may explain the aberrant hypermethylation in some of our patients with inflammatory disease. Finally, the two cohorts had different exclusion criteria regarding previous lung cancer. This may also explain some of the observed differences, although we would have expected more false positive results in the Discovery cohort, which allowed enrolment of participants with a previous lung cancer.

The implementation of CT-based screening on a high-risk population could lead to an increase in redundant, and possibly dangerous, invasive diagnostic procedures. The NLST reported a rate of 1.4% for at least one complication after diagnostic work-up in the CT group [5]. The NELSON trial reported no adverse events at all [6]. According to the NELSON trial, they were restrictive in referring patients for invasive diagnostic work-up, which may have contributed to the low complication rate. However, a meta-analysis of transthoracic biopsy from 2017 found a risk of pneumothorax of 25.3% and 18.8% for core biopsy and fine needle aspiration, respectively [35]. The major complication rates for these biopsy modalities were, respectively, 5.7% and 4.4%. An increase in the number of patients referred for diagnostic work-up would likely increase the number of patients eligible for biopsy—whether transthoracic or by bronchoscopy. In this respect, analyzing bronchial lavage fluid for tumor DNA could be used as a supplementary tool to identify patients with lung cancer in cases when biopsy would entail considerable risk of complications. Blood-based methylation markers would be easier and less invasive to collect, but they may not be as organ specific since many genes are aberrantly methylated in numerous tumors.

The main limitation of the present study was the modest cohort size. A larger cohort is required to be able to include all relevant clinical variables in the prediction model. There were different event rates in the two cohorts, and more lung cancer patients in the Discovery cohort. The participants were recruited consecutively; however, participants with a previous lung cancer diagnosis were not included in the Validation cohort. This was because of the potential risk of increasing the number of false positive results among these participants. The methylated *HOXA9* cut-off and the multiple logistic regression model were developed on data from the Discovery cohort. This could explain some of the differences in the diagnostic properties observed between the two cohorts. We did not use a structured questionnaire or interview guide for registering participant characteristics but relied on the information obtained by the doctor in the medical record. A structured approach would have generated more reliable data. Analysis of tumor DNA in bronchial lavage is a relatively new approach in the diagnosis of lung cancer, and there is no consensus on the best material to use. Sputum [15], pleural effusion [30], bronchoalveolar lavage fluid [31] and bronchial lavage/bronchial wash fluid [17] have all been suggested. We chose to analyze bronchial lavage using only the supernatant, because we aimed to detect cell-free tumor DNA. In future studies it would be relevant to compare the DNA yield and diagnostic accuracy between pellet and supernatant methods.

## 5. Conclusions

In conclusion, we find that methylated *HOXA9* in bronchial lavage holds potential as a supplementary tool in the diagnosis of lung cancer because it has a clinically relevant sensitivity and specificity. Methylated *HOXA9* remained significant when adjusting for age and smoking status in a predictive model. Routine clinical application awaits further validation in a clinical trial.

## Figures and Tables

**Figure 1 cancers-13-04223-f001:**
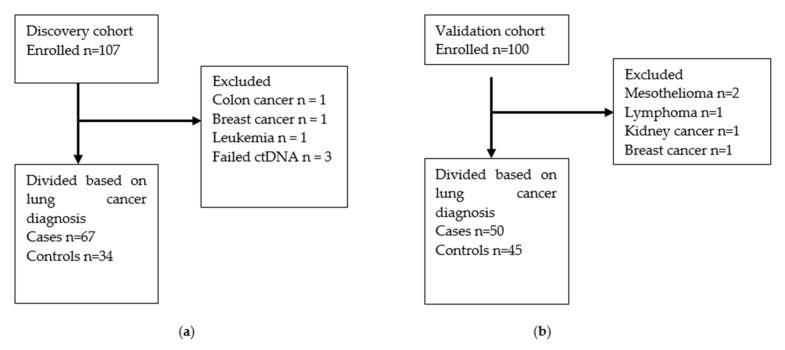
Flow charts illustrating how participants were selected and excluded for (**a**) the Discovery cohort and (**b**) the Validation cohort.

**Figure 2 cancers-13-04223-f002:**
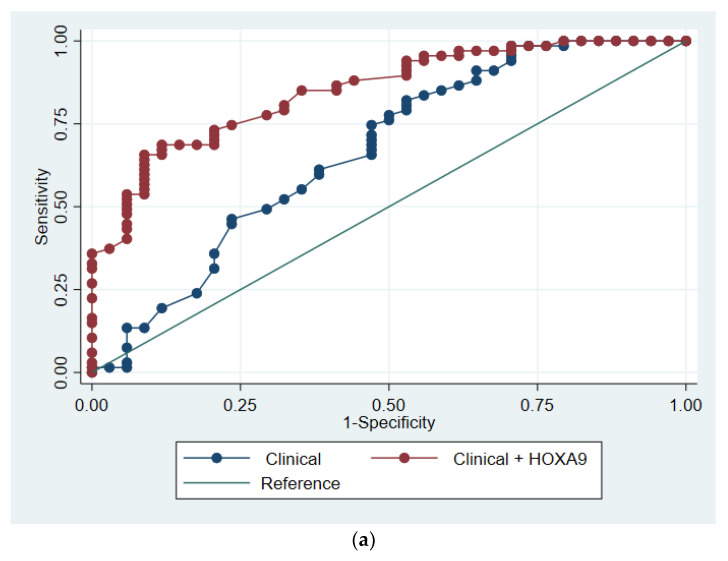
ROC analysis of the multiple logistic regression models based on clinical variables with and without methylated HOXA9. Receiver operating characteristics (ROC) curves illustrating the diagnostic performance of the two logistic regression models in (**a**) the Discovery cohort and (**b**) the Validation cohort. The blue line represents the multiple logistic regression model based on clinical variables, while the red line represents the model including methylated *HOXA9* status.

**Table 1 cancers-13-04223-t001:** Participant characteristics for the Discovery and Validation cohorts, respectively, separated into columns for cases and controls. Continuous variables are reported as median (interquartile range, IQR), and categorical variables are reported as fraction (percentage, %). As a result of rounding, not all categories add up to 100%.

Variable	Discovery Cohort Cases (*n* = 67)	Discovery Cohort Controls (*n* = 34)	Validation Cohort Cases (*n* = 50)	Validation Cohort Controls (*n* = 45)
**Basic information**				
Age, years	73 (65–77)	64 (54–73)	69 (63–77)	63 (56–72)
Sex, male	39/67 (58%)	19/34 (56%)	24/50 (48%)	26/45 (58%)
Employment status, employed	5/65 (8%)	11/29 (38%)	10/47 (21%)	21/39 (54%)
Smoking status:				
Never	7/67 (10%)	6/34 (18%)	2/50 (4%)	13/45 (29%)
Ever	60/67 (90%)	28/34 (82%)	48/50 (96%)	32/45 (71%)
Pack years	40 (20–50)	20 (12–40)	35 (21–50)	15 (0–45)
Performance status:				
0	27/67 (40%)	19/33 (58%)	34/49 (69%)	30/42 (71%)
1	27/67 (40%)	9/33 (27%)	10/49 (20%)	10/42 (24%)
≥2	13/67 (19%)	5/33 (15%)	5/49 (10%)	2/42 (5%)
FEV1, liter	1.87 (1.24–2.44)	2.33 (1.52–2.99)	2.09 (1.54–2.44)	2.54 (1.95–2.85)
**Comorbidity**				
Any comorbidity	58/67 (87%)	28/34 (82%)	43/50 (86%)	37/45 (82%)
Cancer within 5 years	26/67 (39%)	9/34 (26%)	5/50 (10%)	3/45 (7%)
Previous lung cancer	24/67 (36%)	6/34 (18%)	0/50 (0%)	0/45 (0%)

**Table 2 cancers-13-04223-t002:** Tumor characteristics for the Discovery and Validation cohorts, respectively, separated into columns for cases and controls. Stage and histology are only reported for cases. Continuous variables are reported as median (interquartile range, IQR), and categorical variables are reported as fraction (percentage, %). As a result of rounding, not all categories add up to 100%.

	Discovery Cohort		Validation Cohort	
Variable	Cases (*n* = 67)	Controls (*n* = 34)	Cases (*n* = 50)	Controls (*n* = 45)
Tumor on CT scan	60	11	50	28
Largest mean diameter, mm	27 (19–47)	19 (13–22)	42 (23–70)	22 (14–50)
Localization	-	-	-	-
Central	19/60 (28%)	3/11 (27%)	12/50 (24%)	6/28 (21%)
Intermediate	15/60 (22%)	2/11 (18%)	22/50 (44%)	11/28 (39%)
Peripheral	26/60 (39%)	6/11 (55%)	16/50 (32%)	11/28 (39%)
**Confirmed lung cancer**				
Stage		-	-	-
1	18/65 (28%)	-	5/48 (10%)	-
2	7/65 (11%)	-	14/48 (29%)	-
3	16/65 (25%)	-	16/48 (33%)	-
4	24/65 (37%)	-	13/48 (27%)	-
Histology		-		-
Adenocarcinoma	32/67 (48%)	-	34/50 (68%)	-
Squamous cell carcinoma	22/67 (33%)	-	8/50 (16%)	-
Small cell carcinoma	6/67 (9%)	-	3/50 (6%)	-
Other non-small cell lung cancer	7/67 (10%)	-	5/50 (10%)	-

**Table 3 cancers-13-04223-t003:** Methylated *HOXA9* as a diagnostic biomarker for lung cancer reported for the Discovery and Validation cohorts, respectively. *HOXA9*+ indicates detectable methylated *HOXA9* (≥0.13%) in the bronchial lavage sample and hence a positive test. *HOXA9*- indicates a negative test with no detectable methylated *HOXA9* (<0.13%). Sensitivity, specificity, positive predictive value (PPV) and negative predictive value (NPV) are reported with 95% confidence intervals (CI).

	Discovery Cohort			Validation Cohort		
Diagnosis	*HOXA9*+	*HOXA9*−	Total	*HOXA9*+	*HOXA9*−	Total
Lung cancer	49	18	67	40	10	50
No lung cancer	5	29	34	11	34	45
Total	54	47	101	51	44	95
Sensitivity (95% CI)	73.1% (60.9%–83.2%)			80.0% (66.3%–90.0%)		
Specificity (95% CI)	85.3% (68.9%–95.0%)			75.6% (60.5%–87.1%)		
PPV (95% CI)	90.7% (79.7%–96.9%)			78.4% (64.7%–88.7%)		
NPV (95% CI)	61.7% (46.4%–75.5%)			77.3% (62.2%–88.5%)		

**Table 4 cancers-13-04223-t004:** Multiple logistic regression model developed on data from the Discovery cohort and subsequently applied to data from the Validation cohort. The model included data on 101 and 95 participants, respectively, from the Discovery and Validation cohorts. * Statistically significant impact.

	Discovery Cohort			Validation Cohort		
	Cases (*n* = 67)	Controls (*n* = 34)		Cases (*n* = 50)	Controls (*n* = 45)	
Variables	OR (95% CI)	OR (95% CI)	*p*-Value	OR (95% CI)	OR (95% CI)	*p*-Value
Age, years	-	-	-	-	-	-
Crude OR	1 (ref)	1.06 (1.02–1.11)	0.002 *	1 (ref)	1.08 (1.03–1.13)	0.001 *
Adjusted OR	1 (ref)	1.06 (1.01–1.11)	0.023 *	1 (ref)	1.07 (1.01–1.14)	0.020 *
Smoking status, ever	-	-	-	-	-	-
Percentage ever smoked	89.6%	82.4%	0.307	96.0%	71.1%	0.001 *
Crude OR	1 (ref)	1.84 (0.56–5.97)	0.312	1 (ref)	9.75 (2.06–46.14)	0.004 *
Adjusted OR	1 (ref)	1.07 (0.24–4.74)	0.928	1 (ref)	5.20 (0.82–32.95)	0.080
*HOXA9* status	-	-	-	-	-	-
Percentage *HOXA9*+	68.7%	11.8%	<0.001 *	80.0%	24.4%	<0.001 *
Crude OR	1 (ref)	15.80 (5.30–47.06)	<0.001 *	1 (ref)	12.36 (4.68–32.64)	<0.001 *
Adjusted OR	1 (ref)	14.27 (4.62–44.06)	<0.001 *	1 (ref)	11.95 (4.11–34.75)	<0.001 *

## Data Availability

The data presented in this study are available on request from the corresponding author. The data are not publicly available due to ethical considerations.

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
