# Peer review of "Validating Methylated HOXA9 in Bronchial Lavage as a Diagnostic Tool in Patients Suspected of Lung Cancer"

_cancers, 2021, doi:10.3390/cancers13164223_

Round 1
Reviewer 1 Report
In this manuscript, the authors aimed to develop and validate methylated HOXA9 in bronchial lavage as a diagnostic tool in patients suspected of lung cancer. My main concern with this study is that study criteria differed between the Discovery cohort and the Validation cohort. In particular, the Discovery cohort included a non-trivial number of patients with previous lung cancer. More than one-third (24 of 67) of the participants in the Discovery cohort of cases had previous lung cancer. Although the authors acknowledged this fact, it does not justify including them in the analysis. I believe this group of 24 patients with previous cancers should be excluded from the analyses, and that all the tables should be repeated without the 24 patients with previous lung cancers. If the authors feel strongly about including the 24 patients with previous cancers, I think this should the sensitivity analysis but not the main analyses. My other comments are as follows:
Table 1: In the discovery cohort of cases, 26 of 67 patients had cancers within 5 years. Were the 24 previous lung cancer part of this group of 26 patients with cancer within 5 years? This row ‘Cancer within 5 years’ needs to be explained because Figure 1 suggested that patients with other cancers were excluded. Please clarify the definition of performance status and how this was determined.
Table 2: Variable ‘largest diameter’, mm. Is this the largest mean diameter of the tumors?
Table 4: Percentage exposed- I would change this is to percentage ever smoked
Lines 144-145: A volume of 2ml bronchial lavage fluid had the best performance of cell free DNA yield and the least degree of PCR inhibition. Why is a smaller volume ideal? If this is the case, why did you choose a sample volume of 10 ml (lines 114)
Reviewer 2 Report
The authors suggest to include HOXA9 methylation assay as a marker to search in bronchial lavage for lung cancer diagnosis, in addition to clinical parameters, to increase the sensitivity and specificity of the diagnostic procedure.
The work is interesting and well described.
I carry only minor revisions to be applied to the text.
- Some linguistic inaccuracies and the appropriate use of scientific nomenclature throughout the text need further revision.
- line 65: promoter instead of promotor
- lines 65-72: HOXA9 is not the only methylation marker found in the literature for lung cancer. Please give more details why the choice of HOXA9.
- lines 114-115: it is not clear why you report "We chose a sample volume of 10 ml" and then "The recovery of bronchial lavage fluid ranged from 3-7 ml". Please clarify.
- lines 136 and 148: was beta2-microglobulin or instead albumin used to normalize for the total amount of DNA? This passage is not clear.
- line 139: it is not very clear why DNA from lymphocytes is used as a control. Is it perhaps meant to check if there is such an alteration of methylation in the lymphocytes, which could somehow affect the result also in the bronchial fluid? Please explain.
- lines 155-156: "The cut-off was chosen as the level of methylated HOXA9, resulting in the highest number of correctly classified samples". That is, which parameter was used exactly? Usually, either the point corresponding to maximum specificity plus maximum sensitivity, or accuracy, is used. Have you used any of these? Or other?
- the whole paragraph 3.1 is descriptive of the case study, so in my opinion it should be moved to the materials and methods and not described as a result.
- Is it possible to explain the "performance status" in table 1?
- In the discussion, I would emphasize the importance of having found the methylation alteration of HOXA9 in the bronchial fluid, while it would not make the same sense to look for it in the blood, since it would represent a non-specific biomarker, because it is an alteration shared with other tumors.
- lines 239-246: Compared to the panel described by Roncarati et al, why would the use of the biomarker suggested by the authors be better? Or would it be more desirable, for example, to add it to that panel?
- lines 253-254: I suggest to emphasize even more the importance of the functional role of HOXA9 and of the timing of onset of its methylation alteration. It is in fact known that early methylation alterations can provide valuable biomarkers for early diagnosis in non-invasive biological matrices (I recommend to cite ref_A) and even as predictive tumor markers (ref_B and ref_C).
Ref_A: Vega-Benedetti AF, et al. Colorectal Cancer Early Detection in Stool Samples Tracing CpG Islands Methylation Alterations Affecting Gene Expression. Int J Mol Sci. 2020 Jun 24;21(12):4494. doi: 10.3390/ijms21124494
Ref_B: Wong Doo N, Makalic E, Joo JE, Vajdic CM, Schmidt DF, Wong EM, Jung CH, Severi G, Park DJ, Chung J, Baglietto L, Prince HM, Seymour JF, et al. Global measures of peripheral blood-derived DNA methylation as a risk factor in the development of mature B-cell neoplasms. Epigenomics. 2016; 8: 55–66. 10.2217/epi.15.97
- line 264: excellent point; I suggest emphasizing the importance of close monitoring in this type of patient.
- line 286: the number of cases does not appear to be so small, although it is actually a very frequent tumor.
The authors show how the treatment of maternal human milk (HM) with ZnCl2 results in a desired antimicrobial effect, also hypothesizing that the mechanisms affected by the treatment are such as to induce morphological alterations of the bacteria observable by scanning electron microscopy.
The work done is interesting and the paper is well written. It certainly deserves to be published, although, in my opinion, after having clarified and deepened some aspects:
1) the authors use ZnCl2 to release Zn2+ ions in solution. The authors should explain, for example in the materials and methods, the reasons for this choice, given that this compound appears to have risks. As the authors themselves affirm in the final sentence of the manuscript, specific evaluations will be necessary on the possibility of using HM thus treated to feed preterm infants. Furthermore, the question arises whether the bactericidal action, attributed by the authors to the Zn2+ ions (known to have such effects), is not in this case possibly attributable also to the Cl- ions. I therefore propose to also add a paragraph in the discussion in which to suggest that further studies could be conducted with zinc compounds normally used as dietary supplements, such as another inorganic compound (zinc sulfate) or organic compounds (zinc gluconate or zinc citrate).
2) In relation to what just said, it is advisable to extend the discussion to another point that appears to me quite crucial: the enrichment of HM with zinc dietary supplements could be not only fundamental for the bactericidal action described by the authors, but also for the prevention of zinc deficiencies, especially considering the fragility of the target recipients. In fact, it is well known that zinc deficiencies, especially if they occur in the neonatal period associated, are often associated to the predisposition to pathologies such as autoimmunity and autism. It is advisable to mention in this regard the following suggested studies:
- for autoimmunity:
1) Ilonen, J et al. The heterogeneous pathogenesis of type 1 diabetes mellitus. Nat. Rev. Endocrinol. 2019, 15, 635–650.
2) Cannas, D et al. Relevance of Essential Trace Elements in Nutrition and Drinking Water for Human Health and Autoimmune Disease Risk. Nutrients. 2020 Jul 13;12(7):2074.
3) Samuelsson, U et al. Low zinc in drinking water is associated with the risk of type 1 diabetes in children. Pediatr. Diabetes 2011, 12, 156–164.
4) Wessels, I et al. Zinc as a Gatekeeper of Immune Function. Nutrients. 2017 Nov 25;9(12):1286.
- for autism:
1) Masini, E et al. An Overview of the Main Genetic, Epigenetic and Environmental Factors Involved in Autism Spectrum Disorder Focusing on Synaptic Activity. Int J Mol Sci. 2020 Nov 5;21(21):8290.
2) Goyal, DK et al. Zinc Deficiency in Autism: A Controlled Study. Insights Biomed. 2019, 4, 4.
3) Yasuda, H et al. Infantile zinc deficiency: Association with autism spectrum disorders. Sci. Rep. 2011, 1, 129.
4) Frederickson, CJ et al. The neurobiology of zinc in health and disease. Nat. Rev. Neurosci. 2005, 6, 449–462.
5) Vyas, Y et al. Influence of maternal zinc supplementation on the development of autism-associated behavioural and synaptic deficits in o_spring Shank3-knockout mice. Mol. Brain 2020, 13, 1–18.
Author Response
Please see the attachment
(We have only replied until "line 286: the number of cases does not appear to be so small, although it is actually a very frequent tumor." The following text seems to be from another review).

Round 2
Reviewer 1 Report
I am not at all convinced by the authors’ response to the main study limitation, namely, the inclusion of previous lung cancers (24 of 67) in the Discovery cohort. The author’s argument that the inclusion of subjects with previous cancers provides a more realistic picture of the usefulness of the biomarker is not at all convincing. I thought the premise of developing a biomarker is to help alleviate the large number of false positive findings of CT scans. I am assuming the clinical practice of follow-up procedures for patients with a known previous lung cancer would be quite different than that of an individual with no history of lung cancer (or other cancers). In fact, I thought this biomarker will be used mainly for individuals with no previous cancers who had CT scans. If you really want to test this biomarkers in subjects with previous lung cancers, then why did you exclude them in your validation cohort.
More importantly, what is the rationale for not presenting the results since you indicated in your response that you have already performed the statistical analyses on both the entire cohort, and with the patients with previous lung cancer. If both sets of results are presented in the paper, the reader will be able to evaluate the findings and draw his/her conclusions.
Author Response
I am not at all convinced by the authors’ response to the main study limitation, namely, the inclusion of previous lung cancers (24 of 67) in the Discovery cohort. The author’s argument that the inclusion of subjects with previous cancers provides a more realistic picture of the usefulness of the biomarker is not at all convincing. I thought the premise of developing a biomarker is to help alleviate the large number of false positive findings of CT scans. I am assuming the clinical practice of follow-up procedures for patients with a known previous lung cancer would be quite different than that of an individual with no history of lung cancer (or other cancers). In fact, I thought this biomarker will be used mainly for individuals with no previous cancers who had CT scans. If you really want to test this biomarkers in subjects with previous lung cancers, then why did you exclude them in your validation cohort.
More importantly, what is the rationale for not presenting the results since you indicated in your response that you have already performed the statistical analyses on both the entire cohort, and with the patients with previous lung cancer. If both sets of results are presented in the paper, the reader will be able to evaluate the findings and draw his/her conclusions.
Reply: We can follow the reviewer’s argument that the reader should be presented with the analyses from both the entire cohort and with previous lung cancers excluded. We have therefore included these analyses in the Supplementary Materials with the addition of a new Table S2 and adjustments to Tables S3 and Table S5. We have also referenced these additions in the text.